# Identifying chemicals associated with irritable bowel syndrome by integrating a transcriptome-wide association study with chemical-gene-interaction analysis

Xinliang Zhao[1,2☯], Yang Yang[3☯], Zhigang Liu[4*], Ruifang Sun[5*], Aili Suo[1*]

**1** Department of Oncology, The First Affiliated Hospital of Xi'an Jiaotong University, Xi'an, Shaanxi, People's Republic of China, **2** Department of General Surgery, Norinco General Hospital, Xi'an, Shaanxi, People's Republic of China, **3** School of Public Health, Shaanxi University of Chinese Medicine, Xianyang, Shaanxi, People's Republic of China, **4** Department of Thoracic Surgery, Shaanxi Provincial Tumor Hospital, Xi'an Jiaotong University, Xi'an, Shaanxi, People's Republic of China, **5** Department of Pathology, School of Basic Medical Sciences, Health Science Center, Xi'an Jiaotong University, Shaanxi, People's Republic of China

☯ These authors contributed equally to this work.
* lzg1103@163.com (ZGL); ruifang_sun@xjtu.edu.cn (RFS); ailisuo@xjtufh.edu.cn (ALS)

## Abstract

### Background

Irritable bowel syndrome (IBS) is a prominent functional gastrointestinal disorder, yet the precise causes and mechanisms behind it remain largely unclear. Numerous environmental compounds have been associated with the intestinal health of individuals suffering from IBS. This study sought to explore the impact of environmental chemicals on the condition of IBS.

### Methods

We analyzed genome-wide association study (GWAS) data comprising 455,321 individuals of white British descent, among which 28,518 individuals with IBS underwent transcriptome-wide association study (TWAS) analysis. Reference gene expression data were sourced from tissues including the small intestine, transverse colon, sigmoid colon, whole blood, and peripheral blood.

### Results

Gene Ontology and Kyoto Encyclopedia of Genes and Genomes enrichment analyses were conducted utilizing the significant genes identified through TWAS. Additionally, protein-protein interaction network analysis was performed using the STRING database to elucidate the functions of the proteins encoded by these genes. Furthermore, chemical-related gene set enrichment analysis (CGSEA) was employed to explore the associations between environmental chemicals and IBS. In total, TWAS

**Data availability statement:** GWAS summary statistics for irritable bowel syndrome were obtained from Wu et al. (Nat Commun, 2021, 12(1): 1146) and are available at https://cnsgenomics.com/content/data. TWAS analyses were performed using FUSION (http://gusevlab.org/projects/fusion/). Gene expression data (GEO accession: GSE166869) were downloaded from GEO (https://www.ncbi.nlm.nih.gov/geo/query/acc.cgi?acc=GSE166869). PPI information was retrieved from the STRING database (https://cn.string-db.org/). Chemical-gene interaction data were acquired from CTD (http://ctdbase.org/downloads/). All datasets are publicly available.

**Funding:** This study was supported by [Shaanxi Province Natural Science Foundation] in the form of a grant awarded to [ZG L] (2025JC-YBMS-992) and [Shaanxi Province Natural Science Foundation] in the form of a salary for [ZG L]. The specific roles of this author are articulated in the 'author contributions' section. The funders had no role in study design, data collection and analysis, decision to publish, or preparation of the manuscript.

**Competing interests:** The authors have declared that no competing interests exist.

identified 33 significant genes ($P_{FDR} < 0.05$), while CGSEA revealed 112 chemicals significantly correlated with IBS ($P_{FDR} < 0.05$, $|NES| > 1$). Both TWAS (targeting genetic influences) and CGSEA (focusing on environmental influences) were instrumental in pinpointing chemicals associated with IBS.

## Conclusion

The results of this study enhance our comprehension of the genetic and environmental determinants associated with functional gastrointestinal disorders.

## Introduction

Irritable bowel syndrome (IBS) represents a significant functional gastrointestinal disorder, exhibiting a prevalence rate ranging from approximately 7% to 21% [1]. Although IBS usually appears in early adulthood [2], its prevalence has significantly increased among children over the last two decades [3]. There are notable gender differences in IBS prevalence, with women in some Western countries being nearly twice as likely to be diagnosed compared to men; this disparity may be even greater in Asian countries [4]. IBS is mainly characterized by recurring abdominal pain and changes in bowel habits [5]. Various factors may contribute to the development of IBS, including genetic factors, changes in gut bacteria, immune system issues, problems with bile salt metabolism, and irregular serotonin levels [6]. Notably, interactions between the digestive system and the nervous system (the gut-brain axis) are increasingly recognized as a key pathophysiological mechanism, potentially explaining the high comorbidity with psychiatric conditions like anxiety and depression Despite these insights, the precise etiology and pathogenesis of IBS remain largely elusive. The syndrome considerably impairs nutritional absorption, growth, and overall quality of life for those affected [7]. Consequently, it is crucial to elucidate the etiology and associated risk factors of IBS to inform and optimize treatment and prevention strategies.

The management of IBS incorporates both pharmacological and non-pharmacological approaches, with the primary aim of alleviating distressing symptoms [8]. Available pharmacotherapeutics for IBS encompass antispasmodics, laxatives, antidiarrheals, analgesics, and probiotics. Nevertheless, the effectiveness of many of these medications is limited, with several studies indicating that their benefits do not significantly surpass those of a placebo [9].As a result, there has been a growing emphasis on non-pharmacological interventions for IBS in recent years [10]. Current clinical guidelines advocate for the use of soluble fiber to address overall IBS symptoms, while antispasmodic agents are presently available in the United States for symptom management [11]. Although dietary factors can exacerbate IBS, they also play a critical role in symptom relief [12]. Moreover, various environmental compounds are linked to intestinal health in patients with IBS [13]. Research has been conducted to explore the interconnections among complex diseases, genetic factors, and environmental chemicals by evaluating the interactions between genetic

predispositions and environmental influences [14,15]. Therefore, understanding the relationship between environmental chemicals and IBS is vital for both treatment and prevention of the syndrome. Identifying specific chemicals associated with IBS could pave the way for novel preventive approaches based on exposure reduction, opening the possibility of developing preventive strategies based on chemical exposure identification.

Advancements in bioinformatics and high-throughput sequencing technologies have significantly increased the focus on the genetic factors contributing to IBS. Genome-wide association studies (GWAS) have become a key method for investigating the genetic loci linked to complex human diseases and traits. Through these studies, six genetic susceptibility loci for IBS have been identified: *NCAM1, CADM2, PHF2/FAM120A, DOCK9, CKAP2/TPTE2P3*, and *BAG6* [16]. However, GWAS has its limitations, particularly in assessing the risk associated with complex polygenic disorders, as most of the identified single nucleotide polymorphisms (SNPs) are found in non-coding regions of genes [17]. To overcome this issue, transcriptome-wide association studies (TWAS) have been proposed. TWAS combines GWAS findings with expression quantitative trait loci (eQTLs) data to provide a clearer picture of gene-trait associations [18]. In this approach, pre-computed gene expression weights are integrated with SNP data, which enhances our understanding of the genetic architecture that underlies IBS [19].

This study assessed how genetic determinants affect IBS by performing a TWAS using a GWAS dataset. This analysis evaluated gene expression in various tissues, including the small intestine, transverse colon, sigmoid colon, whole blood, and peripheral blood. Furthermore, we conducted a functional analysis of the genes identified through TWAS and identified chemicals associated with IBS. This research improves our understanding of the genetic and environmental factors contributing to gastrointestinal autoimmune disorders.

## Methods

### GWAS data

The summary data for the IBS GWAS were acquired from previously published research [20]. In detail, the study utilized genotype and phenotype data from the European ancestry subset of the UK Biobank (UKB) cohort, which is a population-based volunteer longitudinal cohort of ~500,000 individuals recruited across 22 centers in the United Kingdom. This European ancestry subset included 456,327 individuals, among whom 348,441 were unrelated.

The IBS phenotype was defined based on multi-source data including death registers, self-reported records, hospital admission data, and primary care records (UKB data field: 131639). Initially, 29,524 participants were identified as original IBS cases; after excluding 1,006 participants who also had inflammatory bowel disease (IBD) diagnoses, 28,518 individuals were finally assigned to the IBS case group, with the remaining 426,803 participants in the European ancestry subset serving as controls.

Genotype data of the UKB cohort were imputed using the Haplotype Reference Consortium (HRC) and UK10K as reference samples. Genotype probabilities were transformed into definitive genotypes utilizing PLINK2 (hard-call 0.1). Single nucleotide polymorphisms (SNPs) were excluded from the analysis if they met any of the following criteria: minor allele count < 5, Hardy-Weinberg equilibrium test P-value < $1.0 \times 10^{-5}$, missing genotype rate > 0.05, or imputation accuracy (Info) score < 0.3. Comprehensive details regarding subjects, genotyping procedures, imputation, and quality control measures can be referenced in the original study.

### TWAS analysis

TWAS is a powerful method that integrates gene expression profiles with GWAS summary data. Compared with linkage disequilibrium-based (LD-based) estimates of local genetic correlation, the TWAS approach effectively distinguishes between direct genetic effects on gene expression and indirect LD confounding [21]. In this research, TWAS was conducted utilizing Fusion-based software. Among the most widely used TWAS methods (e.g., PrediXcan, TWAS-Fusion, and

SMR) that interrogate causal relationships between gene expression and complex traits [22]. TWAS-Fusion is favored for its unique advantages in our research: 1) GWAS summary data, without relying on individual-level genotypes; 2) Bayesian sparse linear-mixed models excel at capturing cis-regulatory effects. Briefly, Bayesian sparse linear-mixed models [23] were utilized to calculate SNP expression weights for each gene within its 1-Mb cis-regulatory region. The association between predicted gene expression levels and IBS was then estimated using the formula: $Z = w \cdot Z/(w \cdot [Lw])$ [21], where w represents the SNP weight for gene expression prediction, Z denotes the Z-score from IBS GWAS summary data, and L stands for the SNP correlation matrix (i.e., LD structure). Each genetic feature was defined as a contiguous 100,000 bp genomic segment. For the joint model construction, features with a minimum P-value $\leq 0.05$ were included; features with an $r > 0.9$ were considered redundant (identical) and merged, while those with an $r < 0.008$ were treated as independent to avoid overcounting.

We specifically analyzed gene expression in the small intestine, transverse colon, sigmoid colon, whole blood (comprising blood cells, plasma, and all other components in the blood), and peripheral blood (blood predominantly composed of blood cells), employing pre-calculated functional weights in conjunction with the IBS GWAS summary data to evaluate gene expression through the predictive model available in FUSION. The gene expression weight panel for the aforementioned tissues was downloaded from the FUSION website (http://gusevlab.org/projects/fusion/). A false discovery rate (FDR) correction was applied to all TWAS findings.

## Gene Expression Omnibus (GEO) datasets

Although FUSION enables the mapping of disease-associated loci identified by TWAS to the transcriptional level, a notable limitation exists in its analytical framework: the transcriptomic data utilized for constructing gene expression weights in FUSION are exclusively derived from normal tissues. Such normal tissue-based expression profiles can only partially recapitulate the transcriptional characteristics of disease states, leading to potential inconsistencies between the predicted gene expression patterns and the actual transcriptional landscape in pathological conditions. To address this constraint and improve the reliability of our findings, we employed previously published transcriptomic data retrieved from the GEO database to further filter the candidate genes identified by FUSION, thereby enhancing the credibility and biological relevance of our study outcomes [24].

The GSE166869 dataset was retrieved from the GEO database (https://www.ncbi.nlm.nih.gov/geo/). This study enrolled 26 patients with IBS, including 12 with constipation-predominant IBS (IBS-C) and 14 with diarrhea-predominant IBS (IBS-D), as well as 15 healthy volunteers. Probe-based confocal laser endoscopy was used, and mucosal biopsies of the duodenum and jejunum were obtained for bulk RNA sequencing (RNA-seq). The aim of this study was to identify the small intestinal mechanisms underlying lipid-induced symptoms and rectal hypersensitivity in IBS based on RNA-seq data [25].

## Gene functional analyses

GEO2R [26] was used to find DEGs between diarrhea-irritable bowel syndrome (d-IBS) patients and control groups. Adj.P value < 0.05 and |Log2FC| > 0.5 were considered as differentially expressed genes (DEGs) [27]. Quality control was performed using principal component analysis (PCA) via the prcomp function and ggplot2 to verify sample grouping. The volcano plot and heatmap analysis were done using R packages [28,29].Functional enrichment analysis of DEGs was conducted with the clusterProfiler package for Gene Ontology (GO) and Kyoto Encyclopedia of Genes and Genomes (KEGG) pathways, with significant terms filtered by adjusted P-value < 0.05(Benjamini-Hochberg method for false discovery rate correction) [30].

## Protein-protein interaction (PPI)

PPI networks for TWAS-identified genes were constructed and visualized using Cytoscape software (version 3.9.0 or later), with PPI data retrieved from the STRING database (confidence score $\geq 0.4$ as the cutoff) and visualization

parameters (node size, edge thickness, color gradients) adjusted to distinguish gene/protein nodes and their interaction strengths [31]. Subsequently, the MCODE plugin was applied to the PPI network with optimized core parameters (degree cutoff = 2, node score cutoff = 0.2, k-core = 2, maximum depth from seed = 100) to cluster densely connected sub-networks, and the top 5 modules with the highest MCODE scores were selected as key functional clusters. Meanwhile, the Cyto-Hubba plugin was used to screen hub genes by employing three complementary algorithms: Maximal Clique Centrality (MCC), which accurately identifies functionally critical core hub genes by considering both direct and indirect interactions via clique structures; Edge Percolated Component (EPC), which highlights nodes pivotal to pathway stability by evaluating contributions to network connectivity and robustness; and Degree, a computationally efficient and intuitive method for preliminary screening of hub genes with extensive direct interactions.

### Chemical-related Gene Set Enrichment Analysis (CGSEA)

In this investigation, the chemical gene set enrichment analysis was conducted utilizing the CGSEA tool, which is based on previously published research [32]. The CGSEA tool serves to evaluate the correlations between chemicals and intricate diseases. We employed the Comparative Toxicogenomics Database (CTD) chemical-gene interaction network alongside TWAS expression associations to analyze the statistics related to Irritable Bowel Syndrome (IBS) in order to examine the connections between chemicals and diseases from a genomic functional standpoint. Additionally, we applied weighted Kolmogorov-Smirnov run sum statistics to further investigate the association between chemicals and IBS. The dataset for chemical-associated gene expression annotations utilized in this study was acquired from the CTD (http://ctd-base.org/downloads/), which includes 1,788,149 chemical-gene pair annotation terms, resulting in a compilation of 11,190 chemical-associated genes [33]. To address the issue of multiple testing bias inherent in high-throughput chemical-gene association analysis, we applied FDR correction to the CGSEA results in addition to the P-value filtering. The screening thresholds for significant chemicals were defined as follows: 1) $P\_CGSEA < 0.05$ (primary association signal); 2) $FDR < 0.05$ (corrected for multiple comparisons); 3) $|$normalized enrichment score (NES)$| > 1$ [34].

## Results

### TWAS of IBS

TWAS identified 11,000 unduplicated genes from the GWAS summary data: 2,844, 5,261, 4,829, 4,679, and 2,445 genes for the small intestine, transverse colon, sigmoid colon, whole blood, and peripheral blood, respectively (Fig 1, Supplementary Information S1 File).

### Functional exploration of the TWAS-identified genes associated with IBS

We performed FDR correction on the TWAS-identified genes, yielding 33 significant genes after correction; following the removal of duplicates, 27 genes were retained (Fig 2A). We subjected genes with a TWAS Z value > 0 and Z value < 0 to KEGG and GO enrichment analyses separately (Fig 2B). Results showed that genes with a TWAS Z value > 0 were significantly enriched in pathways including regulation of sodium ion transport, cerebral cortex GABAergic interneuron differentiation, response to histamine, regulation of dopamine receptor signaling pathway, dopamine uptake involved in synaptic transmission, peristalsis, branching morphogenesis of a nerve, negative regulation of ion transport, regulation of cation channel activity, and postsynaptic modulation of chemical synaptic transmission. The activation of sodium ion transport pathways suggests that the upregulation of these genes may be associated with abnormal water-electrolyte metabolism in diarrhea-predominant IBS patients, while the activation of neurotransmitter-related pathways indicates that the upregulation of relevant genes may be linked to abnormalities in neurotransmitter uptake, secretion, and signal transmission in IBS. In contrast, genes with a TWAS Z value < 0 were significantly enriched in pathways such as antigen processing and presentation of endogenous peptide antigen via MHC class I, antigen processing and presentation of endogenous peptide

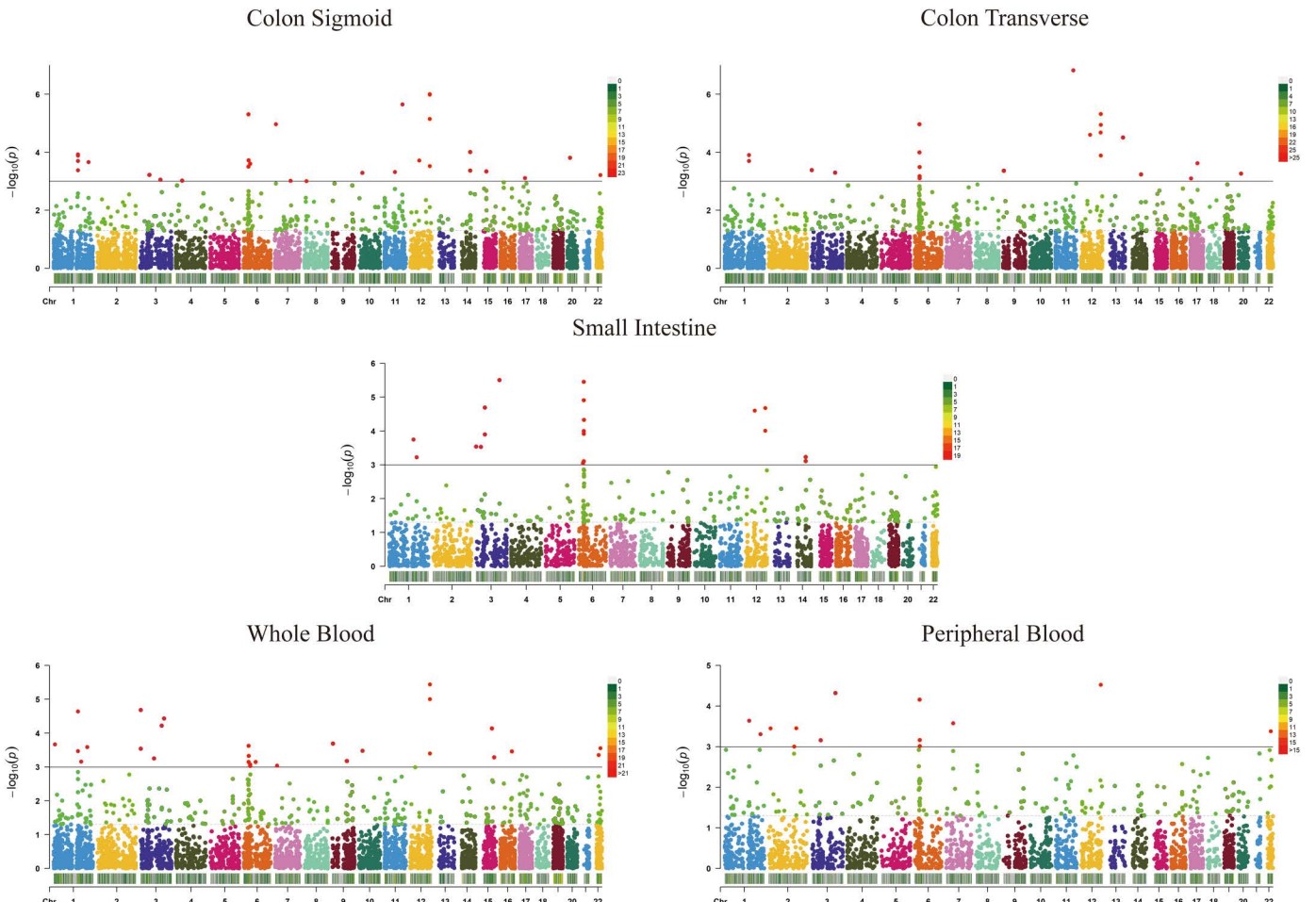

**Fig 1. Manhattan plots of the association results from the whole transcriptome-wide association study.** The horizontal line indicates $P_{TWAS}$, with green dots indicating $P_{TWAS}<0.05$ and red dots indicating $P_{TWAS}<0.01$. Each point represents the genetic predicted expression of a specific gene by the prediction models for small intestine, transverse colon, colon sigmoid, whole blood, and peripheral blood tissues. The x-axis represents the genomic position of the corresponding gene, and the y-axis represents the negative logarithm of the association P-value.

antigen, antigen processing and presentation of endogenous antigen, regulation of integrin activation, positive regulation of macrophage migration, MHC protein complex, MHC protein binding, Natural killer cell mediated cytotoxicity, Allograft rejection, and Autoimmune thyroid disease. The activation of antigen processing and presentation pathways as well as autoimmune-related pathways suggests that the downregulation of these genes may be associated with chronic low-grade inflammation and autoimmune tendency in IBS, while the activation of integrin activation pathways indicates that the downregulation of relevant genes may be related to impaired integrity of the intestinal barrier in IBS.

To identify genes co-expressed in different tissues, we performed an overlap analysis of the 1120 significant TWAS-identified genes ($P_{TWAS}<0.05$). The Venn diagram suggested that there were five significant TWAS-identified genes associated with IBS among the five tissues (Fig 2C, Table 1), including MutS homolog 2 (MSH2), DQ alpha 1 (HLA-DQA1), butyrophilin subfamily 3 member A2 (BTN3A2), transmembrane protein 80 (TMEM80), and cathepsin W (CTSW). Detection of intestinal tissues is notably limited in clinical practice; thus, these five overlapping genes are promising candidates as blood biomarkers that can reflect the gene expression profile of intestinal tissues. Subsequently, we conducted a functional analysis of the genes associated with IBS identified through TWAS. This analysis utilized the Kyoto

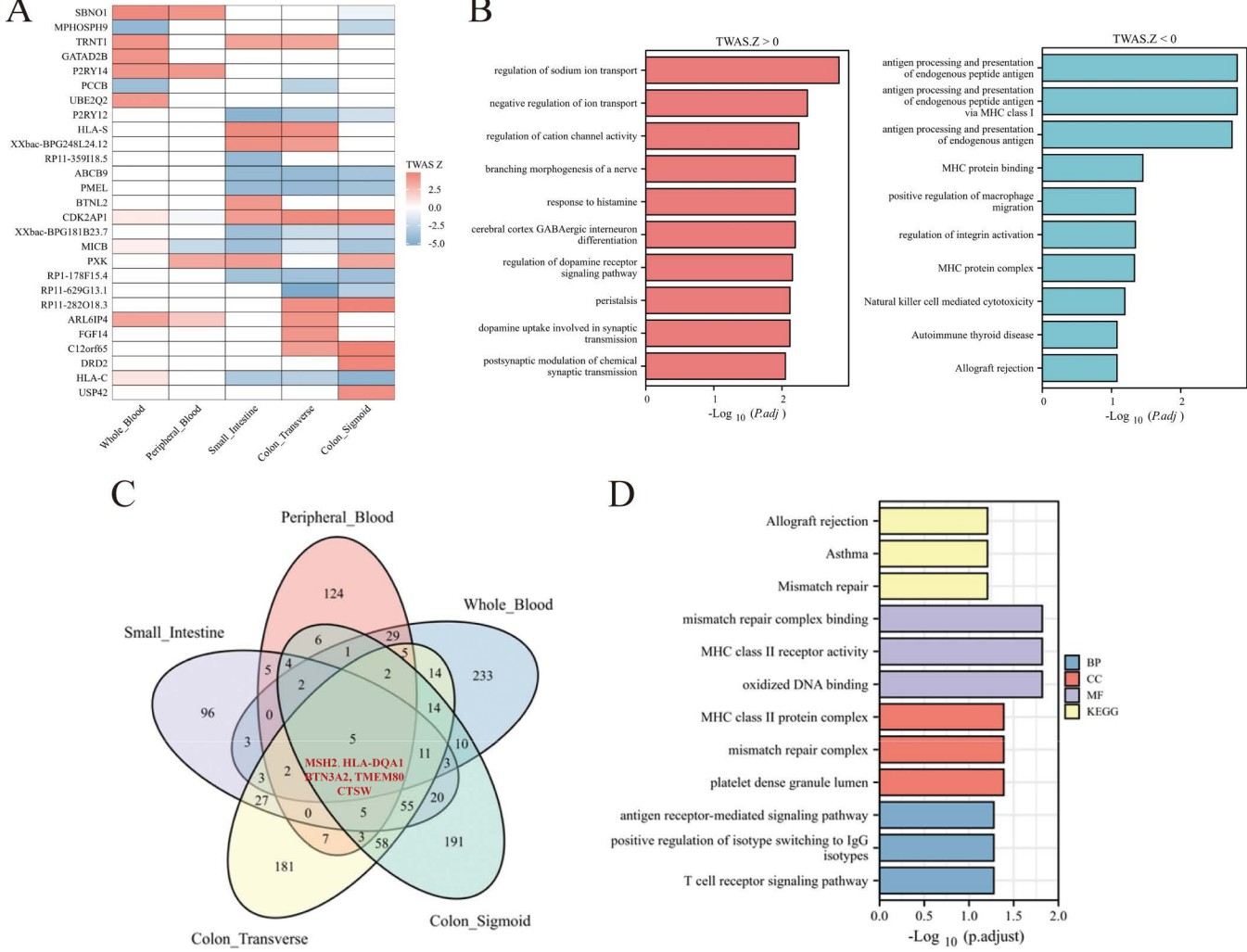

**Fig 2. Exploration of TWAS-identified IBS-susceptible genes.** A: Heatmap of TWAS-significant genes corrected by FDR in all five tissues, shown with TWAS Z value. B: Bar plot of KEGG and GO enrichment of TWAS Z value > 0 genes and TWAS Z value < 0 genes. C: Venn diagram reveals the overlap of TWAS-significant genes in all five tissues. Purple: small intestine; Yellow: transverse colon; Green: sigmoid colon; Red: peripheral blood. D: Bar plot of KEGG and GO enrichment of the 5 overlapping genes. KEGG: Kyoto Encyclopedia of Genes and Genomes; BP: biological processes; CC: cellular components; MF: molecular functions.

Encyclopedia of Genes and Genomes (KEGG) and Gene Ontology (GO) frameworks, and the top three pathways exhibiting significant enrichment from each database are illustrated in Fig 2D. The KEGG pathway analysis highlighted three notably enriched terms linked to asthma, allograft rejection, and mismatch repair mechanisms. In terms of biological processes, three GO terms were significantly enriched, specifically regarding the antigen receptor-mediated signaling pathway, the positive regulation of isotype switching towards IgG subclasses, and the positive regulation of isotype switching to IgG isotypes. Furthermore, three enriched GO terms related to cellular components were identified, including the MHC class II protein complex, the mismatch repair complex, and the lumen of platelet-dense granules. The molecular functions of the enriched GO terms primarily encompassed binding to mismatch repair complexes, MHC class II receptor activity, and the binding of oxidized DNA. These signal pathways suggest that the pathological processes of IBS are mainly characterized by immune-inflammatory imbalance and autoimmune responses.

**Table 1. The TWAS-identified genes for IBS in all five tissues.**

| Gene | Chr | GWAS.ID | $P_{TWAS}$ | | | | |
|------|-----|---------|-----------------|---------------|-----------------|-------------|-----------------|
| | | | **Small Intestine** | **Sigmoid Colon** | **Transverse Colon** | **Whole Blood** | **Peripheral Blood** |
| MSH2 | 2 | rs6749075 | 0.0400 | 0.0300 | 0.0264 | 0.0120 | 0.0400 |
| HLA-DQA1 | 6 | rs532098 | 0.0022 | 0.0170 | 0.0112 | 0.0373 | 0.0265 |
| BTN3A2 | 6 | rs603089 | 0.0009 | 0.0063 | 0.0058 | 0.0023 | 0.0012 |
| TMEM80 | 11 | rs9704354 | 0.0066 | 0.0047 | 0.0100 | 0.0228 | 0.0064 |
| CTSW | 11 | rs603509 | 0.0022 | 0.0019 | 0.0043 | 0.0027 | 0.0026 |

*Nature of values: $P_{TWAS}$ represents the statistical significance P-value for the association between predicted gene expression levels and irritable bowel syndrome (IBS) in transcriptome-wide association study (TWAS). It reflects the strength of the statistical association between gene expression levels and disease phenotypes, with a smaller value indicating a stronger association.

*Tissue definitions: Small intestine, sigmoid colon, and transverse colon represent intestinal mucosal tissues, while whole blood and peripheral blood represent hematopoietic tissues. These tissues cover the core pathological tissues (intestine) relevant to IBS and potential systemic regulatory tissues (blood).

## Tissue-specific functional analysis

We further performed enrichment analysis separately on the differentially expressed genes (DEGs) in the small intestine, colon, and blood tissues (Fig 3). To further capture the relationships between the terms, a subset of enriched terms was selected and rendered as a network plot, where terms with a similarity > 0.3 are connected by edges. The network was visualized using Cytoscape 5, where each node represents an enriched term and is colored first by its cluster ID and then by its p-value. The differentially expressed genes in the three tissues (colon, small intestine, and blood) exhibit distinct tissue-specific pathways, each focusing on their respective physiological roles and exerting unique biological significance in IBS. For the colon, its unique core pathways include VEGFR2 mediated cell proliferation, MHC class II protein complex assembly, Butanoate metabolism, eicosanoid metabolic process, and progesterone metabolic process, with functional focus on intestinal local immune regulation (e.g., MHC class II-mediated antigen presentation in mucosal immunity), utilization of gut microbiota metabolites (e.g., Butanoate, a key energy source for colonic epithelial cells), and intestinal mucosal cell proliferation; biologically, abnormal Butanoate metabolism in the colon may impair intestinal barrier integrity by reducing energy supply to epithelial cells and weakening tight junctions, thereby exacerbating IBS symptoms like abdominal pain and diarrhea, while dysregulated MHC class II protein complex assembly can disrupt antigen presentation to T cells, leading to mucosal immune imbalance and chronic low-grade inflammation (a core pathological feature of IBS), and abnormal VEGFR2-mediated cell proliferation may hinder colonic mucosal repair, further compromising the intestinal barrier. In the small intestine, the unique pathways are mitochondrial tRNA aminoacylation, sulfur compound biosynthetic process, Ovarian steroidogenesis, Primary ovarian insufficiency, and regulation of leukocyte mediated immunity, focusing on energy metabolism (mitochondrial function, as mitochondrial tRNA aminoacylation is critical for mitochondrial protein synthesis), steroid hormone biosynthesis, and local intestinal immune activation (leukocyte-mediated immune responses); in terms of IBS relevance, mitochondrial tRNA aminoacylation abnormalities can reduce energy production in small intestinal epithelial cells, disrupting nutrient absorption and intestinal motility to cause IBS-related malabsorption and irregular bowel habits, dysregulated leukocyte-mediated immunity can trigger local inflammatory responses to accelerate mucosal damage and enhance visceral hypersensitivity (a key driver of IBS abdominal pain), and abnormal Ovarian steroid metabolism may explain the higher prevalence of IBS in women, as sex hormone fluctuations (e.g., during the menstrual cycle) can exacerbate small intestinal dysfunction. As for the blood, its unique core pathways include Cellular responses to stress, Adaptive Immune System, Membrane Trafficking, endocytosis, Metabolism of RNA, and Ebola virus infection in host, with functional emphasis on systemic immune response (regulation by the adaptive immune system), stress adaptation (cellular stress signaling), and subcellular material transport (membrane trafficking and endocytosis); biologically,

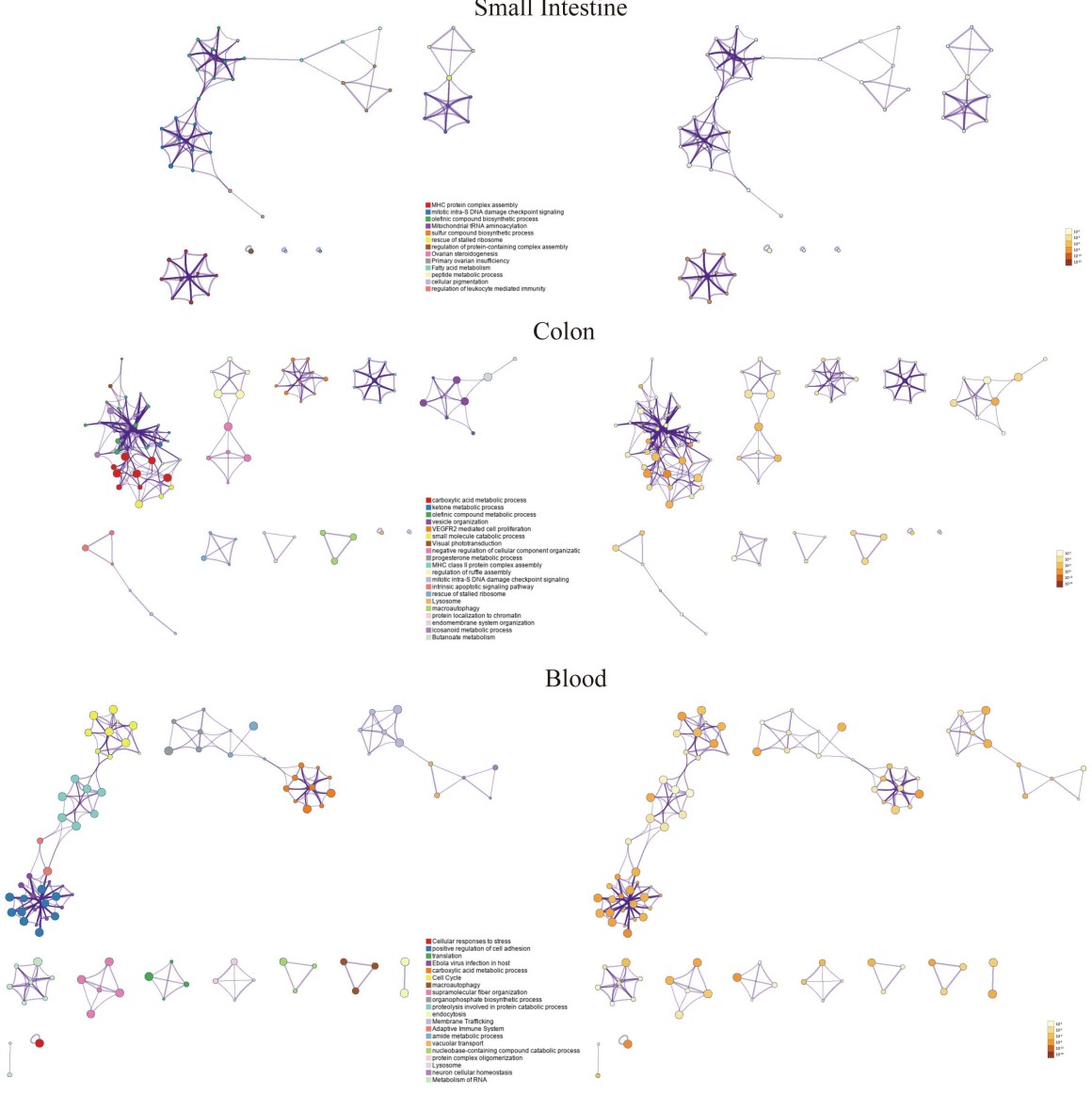

**Fig 3. PPI network and functional exploration of different tissues.**

abnormal cellular responses to stress in the blood highlight the role of the "brain-gut axis" in IBS, as psychological or environmental stressors can trigger systemic stress signaling via the circulatory system to disrupt intestinal motility and visceral sensitivity, explaining stress-induced exacerbation of IBS symptoms (e.g., increased abdominal pain, altered defecation frequency), dysregulated adaptive immunity indicates a systemic inflammatory state in IBS, where circulating immune cells (e.g., T cells, B cells) and inflammatory factors may infiltrate the intestinal mucosa to amplify local inflammation and worsen barrier dysfunction, and impaired membrane trafficking/endocytosis can disrupt the secretion and uptake of neurotransmitters and inflammatory cytokines in the circulatory system, hindering signal communication between blood and intestinal tissues and perpetuating the "intestinal-systemic" pathological loop in IBS.

## Common genes shared by TWAS and mRNA expression profiling

Differential analysis was performed on the small intestinal tissues of GSE166869, resulting in 24 upregulated DEGs and 17 downregulated DEGs. The intersection of DEGs from the small intestinal tissues of GSE166869 and TWAS-identified significant genes in small intestinal tissues was taken; unfortunately, no overlapping genes were obtained (Fig 4A). Differential analysis was conducted on the colon tissues of GSE166869, yielding 616 upregulated DEGs and 625 downregulated DEGs. Taking the intersection of DEGs from the colon tissues of GSE166869 and TWAS-identified significant genes in colon tissues, 24 overlapping genes were acquired (Fig 4B). The details of these 24 overlapping genes are detailed in Table 2, including 13 upregulated DEGs and 11 downregulated DEGs. KEGG and GO enrichment analyses were performed on the upregulated and downregulated DEGs among the overlapping genes, respectively (Fig 4C). The activation of inflammatory mediators and inflammatory pathways suggests that the upregulation of these genes may be associated with excessive inflammatory activation in irritable bowel syndrome (IBS). The activation of Glutathione metabolism and branched-chain amino acid degradation pathways indicates that the upregulation of relevant genes may be related to oxidative stress in the intestinal cells of IBS patients. The downregulation of MHC class II antigen processing and presentation pathways suggests that the downregulation of relevant genes may be associated with immune recognition defects in IBS. The downregulation of pathways related to negative regulation of cell growth indicates that the downregulation of corresponding genes may be linked to abnormal cell proliferation and impaired mucosal repair capacity in IBS.

Notably, the tissue-specific distribution of TWAS genes and GEO-derived DEGs shows strong consistency: colon-related genes (629 TWAS candidate genes, 1229 DEGs) are substantially more abundant than small intestinal-related genes (241 TWAS candidate genes, 41 DEGs) in both datasets. This discrepancy highlights that the pathological changes of IBS are predominantly concentrated in the colon rather than the small intestine, which is consistent with the clinical manifestation of IBS—abnormal intestinal function mainly involving the colonic segment. Further analysis reveals three key implications: first, the quantitative dominance of colon-related genes reflects the higher genetic and transcriptional regulatory activity in colonic tissues during IBS pathogenesis, suggesting that the colon is the core tissue mediating IBS-related genetic effects and transcriptional dysregulation. Second, the lack of overlapping genes between small intestinal TWAS candidates and GEO-derived small intestinal DEGs may be attributed to the relatively milder pathological alterations in the small intestine of IBS patients, leading to weak transcriptional signals that are difficult to capture. Third, the consistent tissue-specific enrichment pattern between TWAS and GEO datasets enhances the reliability of our findings—both genetic association (TWAS) and transcriptional change (DEGs) analyses independently confirm the colon as the primary site of IBS pathology, providing a solid basis for subsequent targeted exploration of colon-specific biomarkers and therapeutic targets. Regarding the numerical difference between tissue-specific gene sets, the large gap between colon and small intestinal genes (both in TWAS and GEO datasets) further reflects the tissue heterogeneity of IBS genetic regulatory networks. Genetic effects and transcriptional responses during disease progression are not uniformly distributed across intestinal segments; instead, they are concentrated in the colon, which may be related to the colon's unique physiological functions (e.g., water absorption, fecal formation, and microbial colonization) that make it more susceptible to environmental and genetic perturbations. This tissue-specific feature also explains the low overlap ratio between small intestinal-related gene sets, while the colon-related gene sets show a relatively meaningful overlap (24/629, 3.82% of colon TWAS genes; 24/1229, 1.95% of colon DEGs)—a ratio that is biologically reasonable for complex diseases, considering the multi-factorial nature of IBS pathogenesis.

## CGSEA of the TWAS-identified genes

We conducted a CGSEA analysis to explore the environmental factors affecting IBS, identifying 1,789 chemicals (Supplementary Information 2 in S1 File), including 112 significant chemicals ($FDR < 0.05$, |normalized enrichment score, |NES| > 1). Table 3 lists the top50 compounds ranked by NES. We classified 112 significant chemicals and mapped them to the genes identified by TWAS (Fig 5A). The classifications include Drugs, Nutrients, Organic Compounds, Pollutants/

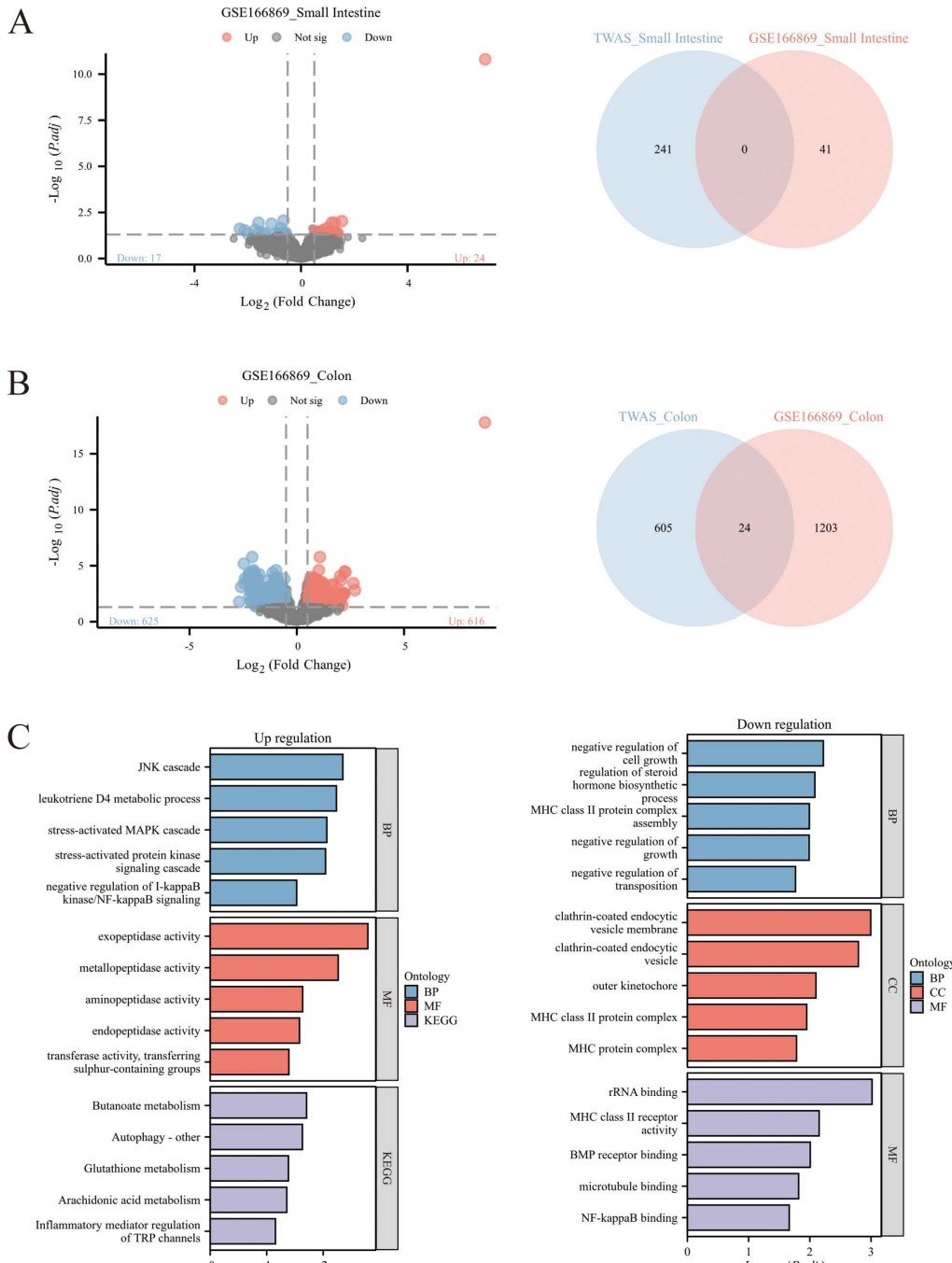

**Fig 4. The analyses of GSE166869 and TWAS-identified genes.** A:volcano plot of small intestinal tissues from GSE166869, and Venn diagram revealing the overlap between differentially expressed genes (DEGs) of small intestinal tissues in GSE166869 and TWAS-significant genes; B: volcano plot of colon tissues from GSE166869, and Venn diagram revealing the overlap between DEGs of colon tissues in GSE166869 and TWAS-significant genes; C: bar charts of KEGG and GO enrichment analyses for upregulated and downregulated genes among the overlapping genes between colon tissue DEGs in GSE166869 and TWAS-significant genes.

**Table 2. Common genes identified by TWAS and GSE166869 DEGs.**

| Gene (up) | adj.P | log2FC | Tissue | Chr | GWAS.ID | Gene (down) | adj.P | log2FC | Tissue | Chr | GWAS.ID |
|---|---|---|---|---|---|---|---|---|---|---|---|
| HLA-DQA2 | 0.032 | −1.20 | Sigmoid Colon | 6 | rs532098 | MYO7A | 0.006 | 1.18 | Sigmoid Colon | 11 | rs1149597 |
| | | | Colon | | | | | | Transverse Colon | | |
| | | | Small Intestine | | | OXCT2 | 0.027 | 1.09 | Sigmoid Colon | 1 | rs2047007 |
| | | | Whole Blood | | | | | | Transverse Colon | | |
| BMP6 | 0.002 | −1.14 | Sigmoid Colon | 6 | rs6597270 | | | | Small Intestine | | |
| | | | Whole Blood | | | TRPV3 | 0.011 | 0.95 | Sigmoid Colon | 17 | rs3826503 |
| EAF2 | 0.003 | −1.02 | Transverse Colon | 3 | rs7340674 | | | | Transverse Colon | | |
| | | | Whole Blood | | | | | | Small Intestine | | |
| | | | Peripheral Blood | | | ATG4B | | | Transverse Colon | 2 | rs7567892 |
| OGFRL1 | 0.017 | −0.98 | Transverse Colon | 6 | rs2273888 | MAPKBP1 | | | Sigmoid Colon | 15 | rs1704401 |
| | | | Whole Blood | | | | | | Whole Blood | 15 | rs12439430 |
| SKA2 | 0.000 | −0.98 | Transverse Colon | 17 | rs7502947 | NPEPL1 | 0.006 | 0.72 | Transverse Colon | 20 | rs6026567 |
| RPF2 | 0.033 | −0.87 | Sigmoid Colon | 6 | rs11751660 | | | | Whole Blood | | |
| | | | Transverse Colon | | | | | | Peripheral Blood | | |
| | | | Whole Blood | | | HYI | 0.021 | 0.70 | Sigmoid Colon | 1 | rs1999595 |
| | | | Peripheral Blood | | | | | | Transverse Colon | | |
| SGIP1 | 0.031 | −0.80 | Transverse Colon | 1 | rs783330 | SH2D3A | 0.047 | 0.68 | Sigmoid Colon | 19 | rs331679 |
| ZNF273 | 0.049 | −0.75 | Sigmoid Colon | 7 | rs35222285 | CDK10 | 0.043 | 0.64 | Sigmoid Colon | 16 | rs7200842 |
| | | | Transverse Colon | | | | | | Transverse Colon | | |
| | | | Small Intestine | | | | | | Small Intestine | | |
| COMMD6 | 0.003 | −0.73 | Sigmoid Colon | 13 | rs9544021 | GGT1 | 0.042 | 0.58 | Transverse Colon | 22 | rs4820600 |
| | | | Whole Blood | | | | | | Small Intestine | | |
| RPL12 | 0.014 | −0.73 | Sigmoid Colon | 9 | rs7847781 | RCE1 | 0.029 | 0.52 | Transverse Colon | 11 | rs7947391 |
| | | | Small Intestine | | | | | | | | |
| | | | Whole Blood | | | | | | | | |
| ULK2 | 0.029 | −0.62 | Sigmoid Colon | 17 | rs8067794 | | | | | | |
| APOBEC3C | 0.044 | −0.58 | Transverse Colon | 22 | rs17304019 | | | | | | |
| | | | Whole Blood | | | | | | | | |
| ZKSCAN8 | 0.019 | −0.53 | Transverse Colon | 6 | rs2799077 | | | | | | |

Heavy Metals/Carcinogens, Inorganic Compounds, and Pesticides. The number and proportion of compounds corresponding to each category are shown in Fig 5B. We further performed PPI (Protein-Protein Interaction) analysis on the TWAS-identified genes and used the MCODE plugin to identify 5 key MCODEs in the PPI network (Fig 5C). Additionally, we analyzed the hub genes in the network using the CytoHubba plugin (Fig 5D). Notably, HLA-DQA1 and HLA-DQA2 are not only hub genes but were also identified in our previous analysis.

## Discussion

The health-related quality of life of IBS patients is significantly reduced, increasing the disease burden [35]. Unlike many other gastrointestinal diseases, IBS is a symptom based diagnosis that presents significant challenges in terms of diagnosis, potential pathophysiology, and management [36]. In addition to gastrointestinal-specific mechanisms, interactions between the digestive system and the nervous system are increasingly recognized among the possible causes of IBS. Therefore, the study of IBS-related gene expression differences can help identify new biomarkers to assist in diagnosis. In

**Table 3. Significantly enriched chemicals identified by CGSEA of IBS.**

| Chemicals | ChemicalName | NES | FDR | Classification | Chemicals | ChemicalName | NES | FDR | Classification |
|---|---|---|---|---|---|---|---|---|---|
| D007099 | Imipramine | 40.97 | 0.014 | Drugs | C013320 | Tris(2-butoxyethyl) phosphate | 10.19 | 0.022 | Organic Compounds |
| C038328 | Bep protocol | 36.14 | 0.014 | Drugs | D006052 | Gold sodium thiomalate | 10.09 | 0.018 | Drugs |
| D008774 | Methylphenidate | 28.82 | 0.021 | Drugs | C063855 | Microcystin rr | 9.91 | 0.024 | Organic Compounds |
| D014800 | Vitallium | 24.76 | 0.030 | Nutrients | D012715 | Sesame oil | 9.83 | 0.037 | Nutrients |
| D005424 | Flecainide | 23.99 | 0.028 | Drugs | C031721 | Naphthalene | 9.51 | 0.026 | Organic Compounds |
| D013750 | Tetrachloroethylene | 19.46 | 0.028 | Organic Compounds | C406224 | Valdecoxib | 9.34 | 0.030 | Drugs |
| D013752 | Tetracycline | 18.78 | 0.012 | Drug | C523799 | Mrk 003 | 9.33 | 0.022 | Drugs |
| D002392 | Catechin | 17.90 | 0.028 | Organic Compounds | C031181 | Phenanthrene | 9.14 | 0.024 | Organic Compounds |
| D012906 | Smoke | 17.44 | 0.024 | Pollutants/ Heavy Metals/ Carcinogen | D015474 | Isotretinoin | 9.03 | 0.045 | Organic Compounds |
| D010426 | Pentosan sulfuric polyester | 16.49 | 0.033 | Drugs | C012568 | Terbufos | 8.47 | 0.025 | Pesticides |
| D019782 | Riluzole | 15.94 | 0.022 | Drugs | C012843 | Cinnamaldehyde | 8.05 | 0.023 | Organic Compounds |
| C030973 | Cupric oxide | 15.87 | 0.026 | Inorganic Compounds | C513428 | Pyraclostrobin | 7.88 | 0.044 | Drug |
| D010433 | Pentylenetetrazole | 13.84 | 0.012 | Drugs | C016766 | Sulforaphane | 7.79 | 0.019 | Organic Compounds |
| D002746 | Chlorpromazine | 13.46 | 0.026 | Drugs | C001277 | Geldanamycin | 7.74 | 0.017 | Drugs |
| C472791 | 3-(4'-hydroxy-3'-adamantylbiphenyl-4-yl) acrylic acid | 13.30 | 0.012 | Drugs | D015741 | Metribolone | 7.65 | 0.037 | Drugs |
| C008435 | Methyldithiocarbamate | 12.82 | 0.017 | Drugs | C568376 | Mt19c compound | 7.58 | 0.033 | Drug |
| C006551 | 2-amino-2-methyl-1-propanol | 12.71 | 0.020 | Organic Compounds | D020245 | P-chloromercuribenzoic acid | 7.55 | 0.020 | Organic Compounds |
| D006997 | Hypochlorous acid | 12.56 | 0.015 | Inorganic Compounds | D014303 | Trinitrotoluene | 7.51 | 0.023 | Organic Compounds |
| C511402 | Grape seed proanthocyanidins | 12.54 | 0.042 | Organic Compounds | D004397 | Fonofos | 7.48 | 0.019 | Pesticides |
| D014873 | Water pollutants | 12.14 | 0.022 | Pollutants/ Heavy Metals/ Carcinogen | D010795 | Phthalic acids | 7.40 | 0.023 | Organic Compounds |
| C487081 | Belinostat | 11.55 | 0.028 | Drugs | C007836 | Geraniol | 7.33 | 0.026 | Organic Compounds |
| C492519 | Diarylpropionitrile | 11.43 | 0.035 | Organic Compounds | C016366 | Dibenzothiophene | 7.33 | 0.036 | Organic Compounds |
| D004026 | Dieldrin | 10.65 | 0.027 | Pesticides | D007530 | Isoflurane | 7.30 | 0.016 | Drugs |
| D007052 | Ibuprofen | 10.49 | 0.016 | Drugs | D010798 | Phycocyanin | 7.24 | 0.046 | Organic Compounds |
| C013698 | Tallow | 10.49 | 0.045 | Organic Compounds | D001556 | Hexachlorocyclohexane | 7.24 | 0.022 | Pesticides |

*The compound IDs and names were retrieved from the Comparative Toxicogenomics Database (CTD, http://ctdbase.org/downloads/), and the PubChem database (https://pubchem.ncbi.nlm.nih.gov/) was used to further verify the compound names. Compound classifications were jointly identified by three researchers with pharmacological or chemical backgrounds. Classifications approved by more than two researchers were included.

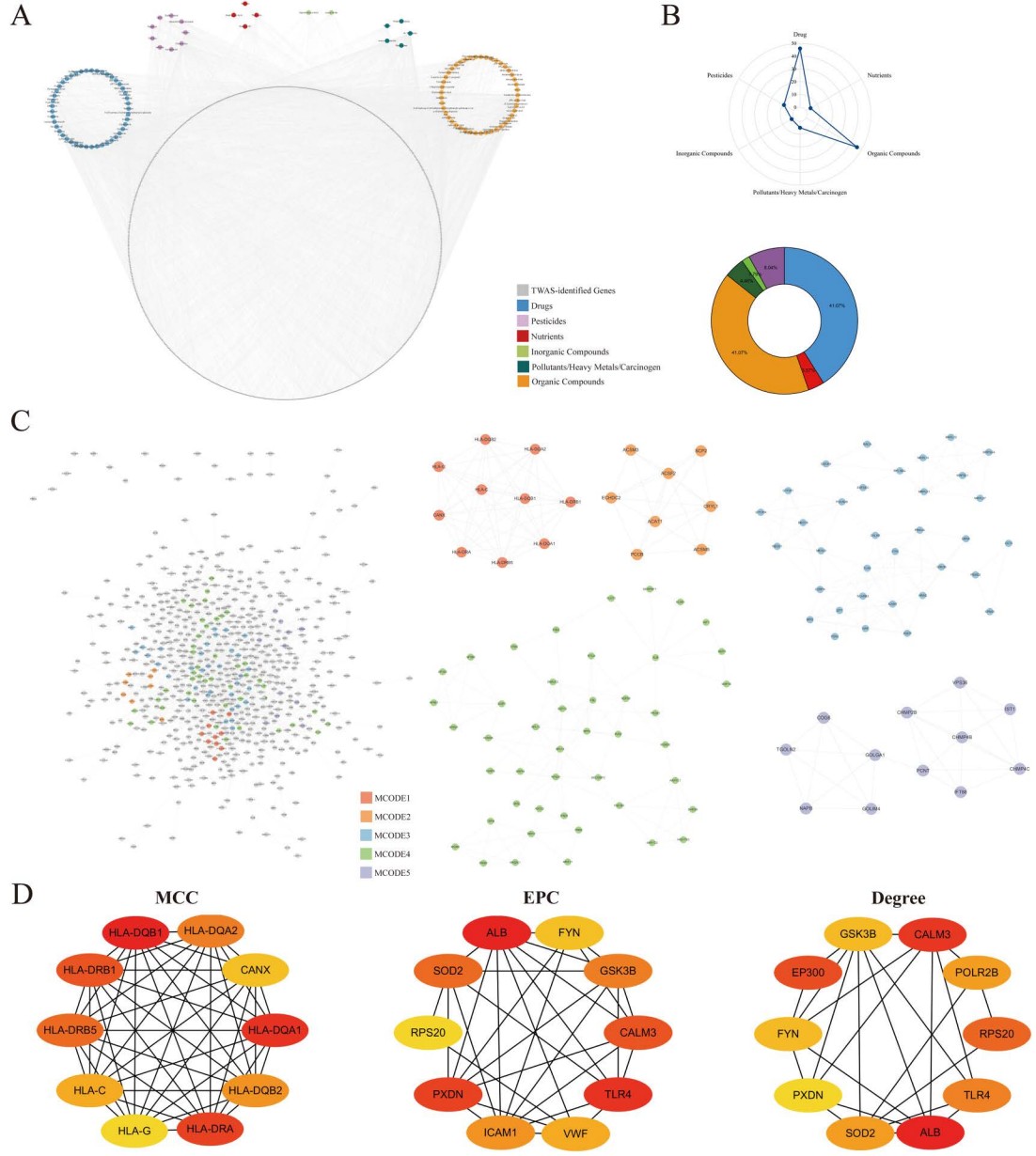

**Fig 5. Network of chemicals and their target genes, based on the TWAS-identified genes. A: Classification of 112 significant chemicals and their mapping to TWAS-identified genes. B: Number and proportion of compounds in each classification. C: PPI network of TWAS-identified genes and top 5 key MCODEs. D: HUB genes in the PPI network.**

addition, studying environmental compounds associated with IBS provides new ideas for identifying risk factors,and opens the possibility of developing preventive strategies based on chemical exposure identification.

We performed TWAS analysis to predict the significant genes expressed in the small intestine, transverse colon, sigmoid colon, whole blood, and peripheral blood. Our results suggest that there are five significant TWAS-identified genes associated with IBS among the five tissues, including *MSH2*, *HLA-DQA1*, *BTN3A2*, *TMEM80*, and *CTSW*. A large-scale epidemiological review [37] has demonstrated a significantly higher incidence of colorectal cancer following an IBS

diagnosis, with an overall risk ratio of approximately 1.5 and a nearly 6.8-fold increase within the first year after diagnosis, indicating that genetic variations in MSH2 (a gene involved in DNA mismatch repair) and related loci may underlie this predisposition. The low level of inflammation underlying the pathogenesis of IBS has received increasing research attention [38]. *HLA-DQA1* (a gene involved in DNA mismatch repair), *BTN3A2* (butyrophilin subfamily 3 member A2, involved in immune regulation), *TMEM80* (transmembrane protein 80), and *CTSW* (cathepsin W, a cysteine protease expressed in lymphocytes) are all associated with immune response. We further performed functional enrichment analysis of these TWAS-identified genes. KEGG and GO enrichment analyses showed that the MHC class II protein complex pathway is the main pathway that plays an important role in IBS. Intestinal epithelial cells express MHC class II molecules during homeostasis and inflammation and can produce different functional outcomes [39]; this is consistent with our findings. We also enriched for sex hormone-related pathways, such as steroid binding, which helps to explain the sex bias in IBS [40]. Neurotransmitter-gut interactions are one of the key regulatory mechanisms underlying the pathogenesis of IBS [41], and our study also identified cellular amide metabolic process pathways, which may involve neurotransmitters such as serotonin (5-HT) and gamma-aminobutyric acid (GABA), known to be critical in gut-brain communication. In this study, we determined some new pathways associated with IBS, such as galactosyltransferase, acetylglucosaminyltransferase, and acetylglucosaminyltransferase activities, showing that the role of gut microbes and their metabolites in IBS is not negligible [42].

IBS is a functional disease [43], thus, avoiding exposure to risk factors and alleviating its symptoms may be more important. We extended the classic GSEA approach to detect the association between environmental chemicals and IBS using CGSEA analysis. We identified various chemicals, including drugs, pesticides, organic compounds, inorganic compounds, nutrients, pollutants, heavy metals, and carcinogens. IBS is often comorbid with psychiatric disorders such as anxiety and depression [44]. The gut-brain axis is a bidirectional neurohumoral integrated communication between the microbiota and autonomic nervous system [45]. Gut-brain interaction disorders are associated with upper and lower gastrointestinal food intake-induced symptoms, which may be an emerging mechanism for studying IBS [46]. Imipramine is the most significant compound for NES obtained in our analysis using CGSEA and is also a commonly used clinical antidepressant [47]. Other antipsychotics, such as methylphenidate, flecainide, riluzole, pentylenetetrazole, and chlorpromazine, were also observed in our analysis. It is noteworthy that while antipsychotics as a class were enriched, not all drugs within this family showed significant associations, suggesting specificity in the mechanisms linking certain psychoactive compounds to IBS pathophysiology.For instance, the differential enrichment of specific tricyclic antidepressants (e.g., imipramine) but not others may reflect their distinct pharmacological profiles, such as varying affinities for neurotransmitter transporters or ion channels that modulate gut-brain signaling. Similarly, among anti-inflammatory agents, only a subset was significantly enriched, pointing to distinct inflammatory pathways being relevant. We have enriched the anti-inflammatory class of drugs, including ibuprofen, valdecoxib, geldanamycin, and metribolone, which support the role played by inflammation in IBS from the perspective of gene-compound interactions. The absence of enrichment for other common anti-inflammatory drugs (e.g., aspirin or certain corticosteroids) may indicate that the association is specific to compounds affecting particular molecular targets (e.g., COX-2 inhibition or HSP90 modulation) rather than a broad class effect. Similarly, among anti-inflammatory agents, only a subset was significantly enriched, pointing to distinct inflammatory pathways being relevant. Another group of compounds of interest is flavonoids such as catechin, grape seed proanthocyanidins, and sesame oil. Research [48] has shown that plant polyphenols are beneficial for IBS by reducing oxidative stress, altering gut permeability, and affecting gut enzyme activity and microbes. A recent study reported gastrointestinal distress in humans after exposure to pesticides, which usually manifests as IBS. Our results also showed that pesticides might be related to IBS, indicating that the effect of pesticides on IBS may be a new direction for future research. However, it is important to note that the associations identified between environmental chemicals and IBS in this study are correlational and do not imply causality; further experimental and epidemiological validation is required to establish causal links.

Our study explored a large number of compounds associated with IBS by analyzing gene-environment interactions, providing novel insights for IBS research. The identification of key genes such as MSH2, HLA-DQA1, and others, along with enriched pathways related to immune function, sex hormones, and neurotransmitter metabolism, advances our understanding of IBS mechanisms. The association of specific chemical classes, including antipsychotics, anti-inflammatories, and flavonoids, with IBS opens new avenues for etiological research and potential preventive strategies. However, this study still has certain limitations. Firstly, the GWAS summary data used in this research is mainly derived from European populations, which means the findings may exhibit varying degrees of differences across different populations. The lack of multi-ethnic validation cohorts (e.g., Asian, African, or American populations) as well as independent clinical cohorts with detailed phenotypic records may restrict the generalizability of our genetic association results. Secondly, the transcriptomic data used to validate the identified IBS-related key genes was primarily obtained from newly diagnosed IBS-D patients who met the Rome III criteria, prospectively recruited at a gastroenterology clinic by Grover, M et al. [25]. The diagnosis of IBS is relatively challenging, resulting in a small number of patients, which may introduce bias into the validation results. Finally, the IBS-related key genes and compounds identified in this study require further validation in subsequent research, including functional experiments to elucidate their biological functions in the pathogenesis of IBS and their potential pathogenic or therapeutic value. Future work should focus on mechanistic studies to compare the actions of enriched versus non-enriched drugs within the same families, longitudinal assessments of chemical exposures in relation to IBS onset, and multi-omics integration to clarify causal pathways.

## Conclusions

This study identified 33 significant genes and 112 environmental chemicals associated with IBS through integrated transcriptomic and chemical-gene interaction analyses. Five key genes (MSH2, HLA-DQA1, BTN3A2, TMEM80, and CTSW) were consistently expressed across all examined tissues, primarily involved in immune-inflammatory pathways and gut-brain interactions. The chemical associations highlight potential environmental risk factors, particularly neuroactive and anti-inflammatory compounds. These findings provide new insights into IBS pathogenesis and identify promising targets for future research on biomarkers and therapeutic strategies.

## Supporting information

**S1 File. Supporting information.** Supplementary information 1: Transcriptome-Wide association study results of irritable bowel syndrome. Supplementary information 2: Chemical-Gene-Interaction analysis resluts of the transcriptome-wide association study identified genes.
(ZIP)

## Author contributions

**Formal analysis:** xinliang zhao.

**Software:** Yang Yang.

**Supervision:** Ruifang Sun, Aili Suo.

**Validation:** xinliang zhao.

**Visualization:** Zhigang Liu.

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
