## [Decision Letter · Decision Letter 0]

9 Oct 2025

Identifying Chemicals Associated with Irritable Bowel Syndrome by Integrating a Transcriptome-Wide Association Study with Chemical-Gene-Interaction Analysis

PLOS ONE

Dear Dr. zhao,

Thank you for submitting your manuscript to PLOS ONE. After careful consideration, we feel that it has merit but does not fully meet PLOS ONE’s publication criteria as it currently stands. Therefore, we invite you to submit a revised version of the manuscript that addresses the points raised during the review process.

We look forward to receiving your revised manuscript.

Kind regards,

Yanggang Hong

Academic Editor

PLOS ONE

Journal Requirements:

“This study was founded by Shaanxi Province Natural Science Foundation (2025JC-YBMS-992), Key Research and Development Program of Shaanxi (2023-YBSF-447), Basic-Clinical Joint & Innovative Project of the First Affiliated Hospital of Xi’an Jiaotong University (2022041).”

Please state what role the funders took in the study.  If the funders had no role, please state: 'The funders had no role in study design, data collection and analysis, decision to publish, or preparation of the manuscript.'

Reviewers' comments:

Reviewer's Responses to Questions

**Comments to the Author**

1. Is the manuscript technically sound, and do the data support the conclusions?

Reviewer #1: Yes

Reviewer #2: Partly

Reviewer #3: Partly

2. Has the statistical analysis been performed appropriately and rigorously?

Reviewer #1: Yes

Reviewer #2: N/A

Reviewer #3: Yes

3. Have the authors made all data underlying the findings in their manuscript fully available?

Reviewer #1: Yes

Reviewer #2: No

Reviewer #3: Yes

4. Is the manuscript presented in an intelligible fashion and written in standard English?

Reviewer #1: Yes

Reviewer #2: No

Reviewer #3: No

Reviewer #1: 1.The abstract (28518 IBS cases) and section 2.1 (455321 total sample) have a logical break in data, and the screening process needs to be clearly explained.

2. The statistical correction method for chemical enrichment analysis (such as whether FDR is used) and the screening threshold basis (incomplete citation of standards with | NES |>1) are not specified.

3.Section 2.5 uses a confidence threshold of 0.15 for STRING database (much lower than the conventional 0.4), without explaining the rationale or conducting sensitivity validation.

4.Section 2.4 does not specify the control group design for differential gene screening (such as the experimental details of EGR3 overexpression group vs control group).

5. The Manhattan plot in Figure 1 does not indicate the corresponding organizational type (A-E) of the subgraph, and readers cannot associate the result description.

Figure 2C Venn diagram does not indicate the names of intersecting genes (only 5 genes are mentioned but not listed in the caption/text).

6.Section 3.5 selects 269 chemicals and only lists the top 50, lacking a classification summary (such as the distribution ratio of drugs/pollutants/heavy metals).

7. Verification queue limitations not declared

Reviewer #2: The study investigates the multifactorial etiology of irritable bowel syndrome (IBS) by integrating genetic and environmental analyses. Using GWAS data from more than 455,000 individuals, including over 28,000 IBS cases, the authors performed a transcriptome-wide association study (TWAS) supported by gene expression data from gastrointestinal and blood tissues. Functional enrichment and protein–protein interaction analyses highlighted relevant biological pathways, while chemical-related gene set enrichment analysis (CGSEA) identified 269 environmental chemicals associated with IBS. In total, 33 significant genes were detected, underscoring potential gene–environment interactions. Overall, the manuscript provides a valuable contribution by combining genetic and environmental perspectives to improve our understanding of IBS.

The background and motivation are well structured, and the availability of a large, previously published GWAS dataset strengthens the study. The methods used for TWAS, enrichment, and protein–protein interaction analysis are appropriate and consistent with similar studies. The discussion acknowledges demographic limitations and contextualizes the findings with relevant literature.

Major Concerns:

-The TWAS analysis requires further clarification. The rationale for using the FUSION model and GWAS summary data to infer gene expression should be explained, and the justification for including the additional GEO dataset is not fully clear.

-The statement regarding “33 significant unduplicated genes” alongside the listing of hundreds of genes per tissue is confusing. It is unclear what “unduplicated” means in this context (e.g., non-redundant across tissues, or differentially expressed?). This needs clearer explanation.

-If all genes are significant after FDR correction, the purpose of Figure 2.A is not obvious. The figure legends also require clarification—should the corrected genes across all five tissues amount to five overlapping genes, or ~1,120 combined? This discrepancy should be addressed.

-While enrichment analysis was performed on the common genes, a comparison of GO/KEGG enrichment for significant genes in each tissue individually would provide additional insights.

-Section 3.3 should specify the selection criteria for “enriched terms” and the types of “interactions” used in STRING. Figure 3, especially 3.A, is crowded and difficult to interpret; presenting results for common differentially expressed genes may improve clarity.

-In Section 3.4, TWAS and DEGs are reported to share 18 genes. It would be useful to contextualize this relative to the 1,120 TWAS genes, and to indicate their tissue distribution. Similarly, whether the five common TWAS genes are represented in the DEG results and how they are expressed should be clarified.

-The reported numbers of unique genes (1101 TWAS vs. 315 DEG) should be corrected and explained. The large differences may reflect tissue- or phenotype-specific factors, which merit discussion.

-Figures 4 and 5 require improvement. Figure 4.B shows a healthy sample clustering close to IBS samples—could this be due to normalization or batch effects? Was this also observed in the clustering analysis (Figure 4.D)? Figure 5 would be more informative if it included network analysis outputs such as hub genes or subgraphs.

-Table 3 could be enhanced by categorizing chemicals (e.g., drugs, nutrients), which would add interpretability.

-In the discussion, while drug families such as antipsychotics and anti-inflammatories are highlighted, the lack of enrichment for other drugs in the same families is not addressed. A mechanistic discussion comparing enriched vs. non-enriched drugs would strengthen the interpretation.

-The conclusion is relatively weak and would benefit from more detail, emphasizing the significance of the findings and potential directions for future work.

-Supplementary Information 1–2 was not available for review and should be provided.

Minor Concerns:

-Several typographical and formatting errors should be corrected. Examples include:

-Section title “2.4. Gene functionalanalyses” → “2.4. Gene functional analyses”

-Section 3.2 “IBSamong” → “3.2. IBS among”

-Section 3.4 “GSE14841DEGs” → “GSE14841 DEGs”

-Figure 4 legend “C: vocanol” → “C: volcano”

-A thorough proofreading will improve readability.

Reviewer #3: This work from Xinliang Zhao et al. presents an interesting and powerful data analysis implemented with the aim of identifying diagnostic markers and improving the mechanistic understanding of IBS, a syndrome whose etiology is complex and poorly understood.

GENERAL COMMENTS AND SCIENTIFIC APPROACH

The work carried out here should be better highlighted in the writing. The manuscript suffers from a general lack of explanation and detail, whether in terms of the methodology used, the description of the results obtained, or the discussion of the scope of the results.

Five tissues were studied here, including three in the intestine and two in the blood. The TWAS study highlights five significant genes common to all five tissues and bases its discussion and conclusions on these five genes. As the authors' approach aims to provide useful discoveries for both the etiological and mechanistic understanding of IBS and for diagnosis, it would be relevant for them to also search separately for markers specific to the intestine or specific to blood. In particular, one can imagine the diagnostic or even prognostic value of blood biomarkers that can thus be accessed non-invasively. In this regard, the authors should clarify for the reader the definition and difference between peripheral blood and whole blood.

METHODS AND DATA

*Explain and justify the choice of datasets used, particularly the GEO expression dataset comprising only 9 samples.

*The 2 Supplementary Information files cited in the manuscript must have a title and be described in the body of the manuscript.

*Why was an analysis performed to “find DEGs between EGR3 overexpression and negative control groups”? What is the link with this study?

RESULTS

*A general lack of explanation in the captions for figures and tables hinders their comprehension:

• Fig. 1: specify that the red dots on the figure correspond to the significant TWAS genes

• Fig. 2 A-B: 33 genes are mentioned in the text but only 27 are shown in the figure

• Fig 5: The title alone is not sufficient; add explanatory details in the caption.

• Table 1: Specify the nature and origin of the values presented.

*Some passages are difficult to understand:

Results 3.1:

The sentence “A total of 33 significant unduplicated genes (PFDR<0.05; Figure 1), including 241, 392, 390, 337, and 200 genes, were identified in these five tissues, respectively (Supplementary Information 1)” is confusing and needs to be clarified. According to Fig. 1, we understand that TWAS identified a total of 33 significant genes (from the 27+21+13+19+27 red dots) in the five tissues. In this case, what do the “241, 392, 390, 337, and 200 genes” correspond to?

Results 3.2:

Why say “33 genes in adipose tissue” about these genes when adipose tissue is not mentioned anywhere else in the manuscript?

For a better understanding, specify that the 5 significant genes identified by TWAS are common to all 5 tissues.

*The presentation of the results needs to be reworked:

The biological and chemical contributions of the study that appear in the results could be presented in a more detailed and attractive way in order to provide a basis for discussion.

Results 3.2: The paragraph on functional results simply copies and lists the enriched functional terms shown in Fig. 2D, whereas a summary presentation would avoid repeating redundant terms between several of the databases used (e.g., mismatch repair, MHC class II). In addition, the term “positive regulation of isotype switching towards IgG subclasses” is cited in the text, although it does not appear in Fig. 2D.

Results 3.3: As mentioned above, here the list of functional clusters is partially copied as is from the (truncated!) sentences in the figure, instead of presenting the main clusters in a summary form.

Results 3.5: The presentation of the chemicals identified by the study is virtually non-existent (4 lines) and is limited to the number of products without any mention of the different categories of products found.

INTRODUCTION AND DISCUSSION

*Introduction:

The introduction lacks important concepts about IBS that are developed in the discussion:

-Interactions between the digestive system and the nervous system among the possible causes of IBS.

-The possibility of developing an approach to prevent IBS based on the identification of chemicals linked to this syndrome.

*The discussion needs to be reworked:

Revise the following sentences (awkward phrasing): “Health-related quality of life has been significantly reduced in patients with IBS, increasing the disease burden [31]. Unlike many other gastrointestinal disorders, IBS is a symptom-based diagnosis that presents significant challenges in terms of diagnosis, underlying pathophysiology, and management [32].”

Expand the text by better showing the links between the results obtained and the existing literature, in order to highlight the new insights provided by the results. In addition, ensure that the main elements of the results included in the Discussion have been previously mentioned in the Results section.

For clarity, the few genes mentioned in the discussion should be cited by their full names and functions, not just their gene symbols. For example, citing “MSH2 and related loci” as a possible gene explaining the increased risk of colorectal cancer in IBS patients without mentioning that this gene is related to the mismatch repair functions found three times in Figure 2D does not help the reader understand the value of this study.

Similarly, it would be interesting for the authors to explain which neurotransmitters could be involved in IBS, rather than limiting themselves to the single sentence “Neurotransmitter-gut interactions are one of the key regulatory mechanisms underlying the pathogenesis of IBS [37], and our study also identified cellular amide metabolic process pathways.” Which neurotransmitters may actually be involved in this metabolic pathway?

The following sentence refers to an interesting result that is not explicitly mentioned in the Results: “We also enriched for sex hormone-related pathways, such as steroid binding, which helps to explain the sex bias in IBS [36].” The same applies to the classes of chemicals identified, which are not described in the results.

It should be remembered that correlation does not imply causality and that the associations found here between chemicals and IBS should be confirmed by other studies

*Conclusion too short:

Either expand and clarify it or delete it and merge it with the discussion.

FORM AND EDITING

*Quality of figures must be improved:

The figures should be enlarged to fill the entire available width to make them more prominent and easier to read.

Figs. 1, 3, and 4 are difficult to read even when zoomed in; enlarge the characters and improve their definition.

Fig. 1: The color code to the right of each plot is not explained in the figure caption and does not seem useful for understanding the figure. Therefore, remove it or explain it if it is justified. The figure could be made more compact by superimposing the plots on two columns (the three intestinal tissues on the left, the two blood types on the right).

Fig. 3: The list of cluster IDs is incomplete because it is truncated on the right.

Table 3: Specify in the legend the source of the reference indicated in the “chemicals” column.

*Review the use of acronyms:

-Be sure to define acronyms before using them:

Paragraph 2.6: “CTD” is used before its full form, which appears later; the same applies to CGSEA: link the acronym to its full form in the first sentence.

-Define them only once:

There is no need to repeat the full form of IBS; the acronym is sufficient in 2.6 and 3.2. The same applies to “normalized enrichment score,” which is defined in Methods; NES is sufficient in 3.5.

*Avoid repetition in the text: there is no need to repeat the data accession number seven times in the caption for Fig. 4, or to repeat the names of the five tissues twice in the caption for Fig. 1.

*English corrections and wording:

Caption Fig. 1: “transverse/sigmoid colon” and not “transverse/sigmoid colon”

Caption Fig. 2: “shown” instead of “showing”

Caption Fig. 2D: add “the” in “Bar plot of KEGG and GO enrichment of the 5 overlapping genes”

Results 3.2: Correct the sentence “This analysis utilized the Kyoto Encyclopedia of Genes and Genomes (KEGG) and Gene Ontology (GO) frameworks, revealing that the top three pathways exhibit significant enrichment, as illustrated in Figure 2D.” “ to ”This analysis utilized the Kyoto Encyclopedia of Genes and Genomes (KEGG) and Gene Ontology (GO) frameworks, and the top three pathways exhibiting significant enrichment from each database are illustrated in Figure 2D."

Results 3.3: Reword: rather than “We used 1,120 TWAS-significant genes for the PPI analysis and successfully transformed 904 protein-coding genes,” the sentence should be something like “From the 1,120 TWAS-significant genes found, the PPI analysis was successfully applied to 904 protein-coding genes.”

*Review the spacing between words throughout the manuscript: there are words that are too close together and spaces that are unnecessary.

*Inconsistent use of upper and lower case letters, particularly in the lists of functional pathways (Fig. 3) and chemicals (Table 3).

**Do you want your identity to be public for this peer review?** For information about this choice, including consent withdrawal, please see our Privacy Policy

Reviewer #1: No

Reviewer #2: No

Reviewer #3: No

---

## [Author Response · Author response to Decision Letter 1]

28 Nov 2025

Reply to Editor and Reviewers:

Thank you very much for giving me the opportunity to revise the manuscript. We appreciate the reviewer’s strict scientific attitude and constructive suggestions. Thank you some much for all your contributaion on my research work. We have included a point by point response in this letter. Besides, we have submitted a revised manuscript.

We have already submitted a document named Response to Reviewers.Please review the reply.

---

## [Decision Letter · Decision Letter 1]

26 Dec 2025

Identifying Chemicals Associated with Irritable Bowel Syndrome by Integrating a Transcriptome-Wide Association Study with Chemical-Gene-Interaction Analysis

PONE-D-25-32341R1

Dear Dr. Zhao,

We’re pleased to inform you that your manuscript has been judged scientifically suitable for publication and will be formally accepted for publication once it meets all outstanding technical requirements.

Kind regards,

Yanggang Hong

Academic Editor

PLOS One

Reviewers' comments:

Reviewer #2: The authors have addressed all of my comments, and I appreciate the time and effort they put into revising the manuscript. The changes have clearly improved the paper, and I thank the authors for their careful and thoughtful responses.

Reviewer #3: The authors have taken the reviewers' comments and requests into account to a large extent. They have repeated and improved certain analyses. The new version provides improved and more comprehensive figures and tables, accompanied by more informative captions. The results were analyzed and interpreted more comprehensively, contributing to the study's greater scientific value. The study now makes a significantly greater contribution to the mechanistic understanding of the pathology under investigation.

---

## [Editor Report · Acceptance letter]

PONE-D-25-32341R1

PLOS One

Dear Dr. zhao,

I'm pleased to inform you that your manuscript has been deemed suitable for publication in PLOS One. Congratulations! Your manuscript is now being handed over to our production team.

Kind regards,

on behalf of

Dr. Yanggang Hong

%CORR_ED_EDITOR_ROLE%

PLOS One